# Performance, Meat Quality and Meat Metabolomics Outcomes: Efficacy of 3-Nitrooxypropanol in Feedlot Beef Cattle Diets

**DOI:** 10.3390/ani14172576

**Published:** 2024-09-04

**Authors:** Cibeli Almeida Pedrini, Fábio Souza Machado, Alexandre Rodrigo Mendes Fernandes, Nara Regina Brandão Cônsolo, Fernanda Maria Marins Ocampos, Luiz Alberto Colnago, Alexandre Perdigão, Victor Valério de Carvalho, Tiago Sabella Acedo, Luis Fernando Monteiro Tamassia, Maik Kindermann, Jefferson Rodrigues Gandra

**Affiliations:** 1Faculty of Agricultural Sciences, Federal University of Grande Dourados, Dourados 79804-970, Brazil; ffabiomachado@hotmail.com (F.S.M.); alexandrefernandes@ufgd.edu.br (A.R.M.F.); 2Faculty of Veterinary Medicine and Animal Science, University of São Paulo, Pirassununga 13635-900, Brazil; nara.consolo@usp.br (N.R.B.C.); fmmocampos@gmail.com (F.M.M.O.); 3EMBRAPA—Brazilian Agricultural Research Company, Instrumentation, São Carlos 13560-970, Brazil; luiz.colnago@embrapa.br; 4Innovation and Applied Science Department, DSM Nutritional Products Brazil S.A., São Paulo 01451-905, Brazil; alexandre.perdigao@dsm-firmenich.com (A.P.); victor.carvalho@dsm-firmenich.com (V.V.d.C.); tiago.acedo@dsm-firmenich.com (T.S.A.); luis.tamassia@dsm-firmenich.com (L.F.M.T.); maik.kindermann@dsm-firmenich.com (M.K.); 5Faculty of Veterinary Medicine, Institute of Humid Tropic Studies, Federal University of the South and Southeast of Pará, Xinguara 68555-970, Brazil

**Keywords:** 3-NOP, methane, greenhouse gases, meat quality, metabolites

## Abstract

**Simple Summary:**

This study was conducted with the primary aim of assessing the dosage of 3-nitrooxypropanol, its effectiveness in methane mitigation performance of feedlot cattle, and its impact on the quality of meat. The results obtained through its application have already demonstrated its efficacy as an additive for methane reduction. Consequently, further research is warranted to confirm its suitability for use in commercial production.

**Abstract:**

30 Nellore animals with an average weight of 407.25 ± 2.04 kg, were distributed in a completely randomized design across the following treatments: 1—Control (without inclusion of 3-NOP); 2—BV75 (inclusion of 3-NOP at 75 mg/kg DM); 3—BV100 (inclusion of 3-NOP at 100 mg/kg DM). No significant effects were observed between treatments on ingestive behavior. However, the notable effect on the BWfinal and ADG of animals supplemented with 3-NOP compared to the control group was measurable. Cattle beef receiving 3-NOP exhibited reduced methane emissions (*p* < 0.0001) for all variables analyzed, resulting in an average decrease of 38.2% in methane emissions compared to the control, along with increased hydrogen emissions (g/day) (*p* < 0.0001). While supplementation with BV100 demonstrated lower methane emission, the performance was lower than BV75 in DMI, BWfinal, ADG, and ADG carcasses. Partial separation of metabolomics observed between groups indicated changes in meat metabolism when comparing the control group with the 3-NOP group, identifying metabolites with a variable importance projection (VIP) score > 1. In conclusion, supplementation with 3-NOP effectively reduced methane emissions and did not negatively influence animal performance.

## 1. Introduction

Global meat production has been steadily increasing to meet growing global demands. However, this surge in production has been accompanied by environmental concerns. According to FAO [1], greenhouse gas (GHG) emissions within agricultural properties, linked to both crops and livestock, experienced a 13% growth between 2000 and 2020. Notably, enteric methane (CH_4_) is a potent gas with a global warming potential 28 times greater than that of carbon dioxide (CO_2_) over 100 years [2]. This methane is a byproduct of ruminal fermentation and serves as a pathway for the elimination of metabolic hydrogen produced by microbial metabolism [3], nevertheless, the emission of enteric methane results in energy losses, as a portion of ingested energy is not assimilated but instead released in the form of CH_4_ [4]. The relative contribution of enteric fermentation emissions to total agricultural GHG emissions varies by region, contingent upon the structure of agricultural production and the type of livestock production system [5].

The utilization of more intensive systems, such as feedlots, incorporating diets rich in grains, has the potential to favorably impact the reduction of methane emissions [6]. However, there is a recognized need for additional tools that can efficiently contribute to gas mitigation. Interventions aimed at reducing emissions are primarily rooted in technologies and practices that enhance production efficiency at both the animal and herd levels [4]; consequently, commercial feed additives are being developed to significantly diminish methane loss. It is worth noting that, given the global population of approximately 1.5 million cattle, the application of such additives becomes particularly relevant in intensive systems where cattle are subjected to controlled diets [7].

Several products are currently undergoing testing as additives to reduce greenhouse gas emissions in animal production, with emphasis on an inhibitor known as 3-nitrooxypropanol (3-NOP; Bovaer^®^, DSM Nutritional Products Ltd.), which stands out due to promising results obtained both in laboratory studies and subsequent field data. In experiments utilizing pure cultures, 3-NOP has been demonstrated to inhibit the growth of methanogenic archaea at concentrations that do not adversely affect the growth of non-methanogenic bacteria in the rumen [8]. This compound has been experimentally tested in various ruminant species, including dairy cows, beef cattle, and sheep at different inclusion levels. Overall, this molecule has proven capable of efficiently reducing enteric methane (CH_4_) emissions [3,5,9,10,11,12,13,14]. Structurally, 3-NOP is an analog of methyl coenzyme M reductase (MCR) and functions as a competitive inhibitor. It selectively binds to MCR, temporarily inactivating the enzyme and facilitating the oxidation of the catalytic nickel ion in co-factor F420 from Ni^+^ to Ni^2+^ [2]. Romero-Perez et al. [15] employed a dosage of 2 g 3-NOP/animal/day in the diets of fistulated beef animals, observing a significant decrease of 59.16% on daily methane (CH_4_) production per animal when 3-NOP was administered. Importantly, this effect was achieved without influencing dry matter intake (DMI) and with a reduction in the loss of gross energy, which would otherwise contribute to CH_4_ production. Vyas et al. [16] observed that in cattle fed a finishing diet, a dose of 200 mg/kg of 3-NOP reduced total enteric CH_4_ emissions, while only a numerical reduction was observed when supplemented with 100 mg/kg of 3-NOP, in comparison with the control diet.

In the context of beef cattle in a commercial feedlot, supplementation with 3-NOP tended to reduce DMI using doses of 150, 175, and 200 mg/kg DM of 3-NOP; however, it improved the gain:feed (G:A) ratio by 2.5% and resulted in a reduction in methane production (g/kg DMI) of 17.2, 25.7, and 21.3% for low, medium, and high doses of 3-NOP, respectively, with an overall decrease of 21.7% [17]. Araújo et al. [14] found that Nellore cattle in a feedlot system for finishing, supplemented with 100 mg/kg DM and 150 mg/kg DM of 3-NOP, showed no significant change in DMI compared to the control, however, the efficiency of daily carcass gain (ADG carcass) increased, regardless of the dose tested, dietary supplementation with 3-NOP decreased CH_4_ emissions by 49.3% (g/d). 

Dijkstra et al. [18] conducted a meta-analysis suggesting that the effects of 3-NOP often depend on the supplementation dose, dietary fiber content, and type of livestock; they found evidence that the effectiveness of 3-NOP in mitigating CH_4_ emission is positively associated with the dose of 3-NOP and negatively associated with dietary fiber content. Increases in the concentrations of neutral detergent fiber (NDF) and crude fat above the average in the database reduced the effectiveness of 3-NOP (at a constant given dosage) in mitigating methane production and yield. Conversely, increases in starch content increase the effectiveness of 3-NOP in mitigating the methane yield [19].

Studies on the use of 3-NOP in the nutrition of beef cattle have shown promising results, indicating a reduction in methane emissions and improvements in feed efficiency and overall performance [14,17]. However, no other studies have been identified that reported whether and how muscle deposition can be influenced by changes in the microbiota and ruminal metabolism with the use of this additive. Therefore, novel analytical methods can be employed to better understand how 3-NOP can have effects on meat quality. 

Despite extensive research, there remains interest in determining the optimal doses of 3-NOP for CH_4_ emissions, balancing efficiency in mitigation without compromising animal performance. Therefore, this work seeks to find not only this balance but also a dose that is viable for confinements, testing doses to verify whether methane reduction occurs efficiently, and without affecting the quality of the animals’ meat.

In light of these considerations, the objective of the present study was to assess the effects of 3-Nitrooxypropanol (3-NOP, Bovaer^®^) at different levels in the diet of feedlot cattle using two dosages. The aim was to use a dosage with results already found under some evaluated parameters, compare it with a lower dose, and measure its efficiency in beef cattle feedlots. In this way, to analyze the responses of the doses used on gas emissions, specifically methane. Additionally, the aim was to investigate the impact on ingestive behavior, animal performance, meat quality, and metabolite production.

## 2. Materials and Methods

### 2.1. Location, Animals and Diet

The experiment was conducted at the DSM Experimental Center, situated in Fazenda Caçadinha, in the municipality of Rio Brilhante, MS, Brazil. The study spanned from September to December 2022 and carried out following approval by the ethics committee, with protocol number 008/22 BR 220701.

A total of 30 Nellore animals, with an average weight of 407.25 ± 2.04 kg, were utilized and distributed in a completely randomized design across the following treatments: 1—CON (without inclusion of 3-NOP); 2—BV75 (inclusion of 3-NOP at 75 mg/kg DM); 3—BV100 (inclusion of 3-NOP at 100 mg/kg DM). The additive 3-NOP (Bovaer, DSM Nutritional Products, Kaiseraugst, Switzerland) was supplied and incorporated into the diet before distribution in the animals’ trough. Diets were balanced in accordance with the RLM and NRC 2020, ensuring that they were isonitrogenous and isoenergetic. The animals went through two adaptation periods to adapt to the finishing diet, starting with an intake of 1.5% of BW, and were adjusted daily based on trough readings, allowing the animals to gradually increase their intake. The animals were kept in feedlot for a duration of 86 days and were subsequently slaughtered.

### 2.2. Intake and Performance

The animals were kept in a single pen feedlot area of 24 m^2^/animal. The pen was equipped with 6 electronic troughs from Intergado (AF 1000). The daily intake of dry matter and nutrients was measured by feeders supported on load cells, allowing for electronic recording of the food consumed by each animal. Feed and leftovers sample were collected on a monthly basis and stored in a freezer for subsequent analysis. The analysis included measurements of dry matter (DM; method 930.15), organic matter (OM; calculated as DM minus ash), crude protein (CP; method 984.13, N × 6.25), ether extract (EE; method 920.39), and lignin (method 973.18) following the procedures outlined by AOAC [20]. Additionally, the content of neutral detergent fiber (NDF) and acid detergent fiber (ADF) were measured according to Van Soest et al. [21]. The starch content of the samples was determined by spectrophotometry after enzymatic degradation using Amyloglucosidase AMG 300L (Novozymes, Bagsvaerd, Denmark), as described by Bach Knudsen [22] (Table 1).

The animals’ weights were recorded daily using a precision livestock platform (BOSCH^®^). This platform facilitated weighing by allowing the animals to pass through it, with frequent monitoring for weight measurement. Daily weight gain was calculated, and the average daily gain (ADG) for each animal during the observed period was determined by simple polynomial regression equations using PROC REG in SAS 9.4. The growth curve for each animal was based on the live weight averages. The slope of these equations was generated for each animal based on the live weight averages. The slope of these equations could be observed individually per animal, per treatment, and by obtaining the general average, it accurately represented the total weight gain during the observed period, thus eliminating possible errors attributable to timing or weighing inaccuracies. To determine the initial carcass weight, the prediction equations described in Bel Bianco Benedeti et al. [23] were used, and the average carcass gain (ACG) was calculated using simple polynomial regression equations using PROC REG in SAS 9.4.

### 2.3. Ingestive Behavior

Animal behavior was gathered on a daily basis throughout the 86-day experimental period, individually from the 30 animals subjected to various experimental treatments. Data collection was facilitated using Intergado’s automated online system. The variables scrutinized included intake time (min/day), frequency of visits to the trough (n), and dry matter intake (DMI; measured in min/kg and kg/trough visit). Individual and daily data were subsequently aggregated, and weekly averages were computed for the purpose of statistical analysis. The animals’ DMI was measured only by Intergado’s feeding troughs; the intake of pelleted feed offered by GreenFeed (C-Lock Inc., Rapid City, SD, USA) was not evaluated.

### 2.4. Methane Emission 

Methane gas (CH_4_) production was quantified utilizing the GreenFeed apparatus, which is capable of concurrently measuring carbon dioxide (CO_2_) and hydrogen (H_2_) emissions from individual animals. The collected data were amalgamated to derive emissions data at the herd level, calculating averages for the entire group. The system is configured to dispense a small amount of attractive pelleted bait, which is available to the animals 24 h a day. In this way, animals can access the device at any time, and the number of accesses per animal (drops) is not limited, encouraging them to engage with the device multiple times throughout the day. Gas emission data are meticulously recorded and processed, enabling users to conveniently access summary reports of calculated flows.

The GreenFeed operation commences when an animal inserts its head into a hood. An RFID reader identified the animal’s ear tag, triggering the sampling process when the animal’s head (located by an infrared sensor) was correctly positioned within the hood, and a predetermined time phase elapsed since the last methane measurement for that animal [24]. GreenFeed is programmed using software (C-Lock Inc., Rapid City, SD, USA), facilitating seamless control and monitoring, even over extended distances, through internet connectivity.

### 2.5. Slaughter and Meat Quality

After 86 days of feeding, individual animal weights were recorded using a mechanical scale to obtain their live weights. Following this initial weighing, a 12-h fasting period was imposed, during which the animals were restricted to a water-only diet. Subsequently, a second weighing was conducted after the fasting period to determine the empty weight. The animals were then transported approximately 220 km to a commercial slaughterhouse (JBS S/A—Campo Grande, MS, Brazil) and subsequently processed. Upon reaching the slaughterhouse, the carcasses were longitudinally halved, with each half carcasses identified and weighed. The half carcasses were stored in a cold room at 4 °C for approximately 24 h. Following the cooling period, a section of the Longissimus muscle between the 11th and 13th ribs of each left half carcass was sampled. Two slices of muscle, approximately 2.5 cm thick, were extracted from each animal for the determination of physicochemical parameters in the Longissimus muscle. pH determination was conducted on thawed samples using a calibrated portable digital pH meter (Testo 205, Testo, Lenzkirch, Germany) inserted into the muscle. Color analysis was performed using a previously calibrated digital colorimeter (Chroma Meter CR-400, Konika Minolta Co., Osaka, Japan), measuring luminosity (L*), red (a*), and yellow (b*) in the meat. For the water retention capacity (WHC), the compression method described by Cañeque and Sañudo [25] was employed. In this approach, a sample weighing approximately 2 g was subjected to a force of 2250 kg for 5 min. The WHC result was calculated as the difference between the initial and final weights and was expressed as a percentage. Cooking weight loss (CL) was determined using the methodology described by Osório et al. [26]. Meat samples were roasted in a preheated electric oven at 170 °C until the internal temperature reached 70 °C. The weights of the samples before and after cooking were utilized to calculate the total loss (%). Furthermore, the samples used for CL analysis were employed for shear force (SF) analysis of the cooked samples. Longitudinal strips of muscle fibers were extracted using a cylindrical steel sampler, and these samples were then inserted into the device coupled to a Warner Bratzler blade of 1 mm (TA-XT Plus, Stable Micro Systems Ltd., Godalming, UK) to determine the force required to cut each cylinder transversely. The average force exerted to cut the cylinders was calculated and expressed in kilogram-force (kgf) [26]. The dry matter (DM) content was determined in an oven at 55 °C, since there was water reabsorption in the samples submitted to laboratory analysis, and the obtained results were adjusted for total dry matter. To determine the levels of ether extract (EE), after defrosting the samples and removing the fat present in the meat and connective tissue, the samples were pre-dried in an oven with forced air circulation at 55 °C for 72 h, and at the end of pre-drying, crushed by a food processor to obtain a homogeneous mass. Subsequently, EE was determined by extraction in a Soxhlet apparatus using an organic solvent; and minerals, by burning the already dried and crushed material in a muffle furnace at 600 °C for 16 h [27].

### 2.6. Meat Fatty Acid Profile

To determine the fatty acid (FA) profile, the samples were freeze-dried for 72 h and then ground using a processor. The total lipids were extracted according to Bligh and Dyer [28]. Subsequently, 60 mg of the extracted lipid fraction was weighed and subjected to methylation according to Maia and Rodriguez-Amaya [29], aiming to prepare it for analysis by gas chromatography.

The analysis of fatty acid methyl esters was carried out on a gas chromatograph, equipped with a flame ionization detector, “Split/splitless” injector, fused silica capillary column containing polyethylene glycol as stationary phase (DB-Wax, 30 m × 0.25 mm, J&W Scientific, Folsom, CA, USA), under the following chromatographic conditions: injector temperature 250 °C; column temperature 180 °C for 20 min, programmed at 2 °C per minute until 220 °C; detector temperature 260 °C, hydrogen carrier gas at a flow rate of 1.0 mL/min, nitrogen make-up gas at 20 mL/min, and an injection volume of 1 μL. To identify fatty acids, retention times were compared with those of methyl ester standards (Sigma-Aldrich, Burlington, MA, USA), while quantification was carried out by area normalization, expressing the result as a percentage of the area of each acid over the total area of fatty acids. Quantification was performed using internal standardization [30] and expressed in (g/100 g of fatty acids). The methods used are described in the AOAC [20] (method 996.06) and Choi et al. [31].

The activities of the enzymes 9 desaturases and elongate were determined according to Malau-Aduli et al. [32], Kazala et al. [33], and Pitchford et al. [34], utilizing mathematical indices. The atherogenicity index was calculated based on the proposal by Ulbricht and Southgate [35] as an indicator of the risk of cardiovascular disease. The calculations were performed as follows: 

Δ9 desaturase 16: 100 [(C16:1cis9)/(C16:1cis9 + C16:0)]

Δ9 desaturase 18: 100 [(C18:1cis9)/(C18:1cis9 + C18:0)]

Elongase: 100 [(C18:0 + C18:1cis9)/(C16:0 + C16:1cis9 + C18:0 + C18:1cis9)]

Atherogenicity: [C12:0 + 4(C14:0) + C16:0]/(AGS + AGP).

### 2.7. Metabolomic Analyzes

To carry out metabolomic analyses, muscle samples were collected during slaughter for subsequent extraction of metabolites, NMR spectroscopy, spectral processing, and finally, metabolite identification and quantitation. This analysis aims to identify and quantify the metabolites present in the tissue studied, thus understanding the pathways and changes in metabolism. During the slaughter process, the tissue that would be collected was identified on the slaughterhouse’s slaughter line by a person responsible for collection. During the slaughter line, the cleaned and weighed carcasses went to a cold room at 4 °C for approximately 24 h, but before entering the cooling chamber, a section of the Longissimus muscle between the 11th and 13th ribs of each left half carcass was collected. These muscle samples were also processed, identified and inserted into “Basic” model cryogenic tubes with an external lid, and stored in liquid nitrogen, kept in a 20-L nitrogen cylinder until being transported to the FMVZ laboratory at the University of São Paulo, USP, Pirassununga, SP, where they were transferred to a −80 °C freezer for subsequent metabolomic analysis.

#### 2.7.1. Extraction of Polar Metabolites 

The extraction of polar metabolites was performed using organic solvents, following the procedure described by Antonelo et al. [36] and Cônsolo et al. [37]. In brief, 0.5 g of frozen meat was macerated and homogenized using an ultra-turrax^®^ (T 25 digital, IKA, Campinas, SP, Brazil). Metabolites were extracted using a mixture of cold methanol/chloroform/water (2:2:1 *v*/*v*) by vortexing for 1 min. Samples were then stored on ice for 15 min and centrifuged for 15 min at 10,000× *g* and 4 °C. The supernatants were carefully transferred to Eppendorf tubes and freeze-dried (Itasul Import and Instrumental Technical Ltd.a, Porto Alegre, RS, Brazil). The samples were transported to Embrapa Instrumentação for metabolomic analyses. On the day of analysis, the remaining residues were replenished in 600 µL of the NMR solution and 60 µL of an internal standard solution, vortexed, and spun for one minute. Supernatant samples (600 μL) were then transferred to standard 5 mm × 178 mm thin-walled NMR tubes (VWR International, Radnor, PA, USA).

#### 2.7.2. NMR Spectroscopy and Spectral Processing

Spectra acquisition and processing followed procedures similar to those described by Cônsolo et al. [37,38]; the one-dimensional (1D) ^1^H NMR spectra were acquired at 300 K on a Bruker Avance 14.1 T spectrometer (Bruker Corporation, Karlsruhe, Baden-Württemberg, Germany) at 600.13 MHz for ^1^H, through a BBO 5 mm probe, D2O was used as a lock solvent, and DSS served as the chemical shift reference for ^1^H. Standard one-dimensional (1D) proton NMR spectra were acquired using a single 90° pulse experiment, and each spectrum was the sum of 64 FIDs.

Water suppression was achieved using the Bruker “zgesgp” pulse sequence (excitation sculpting with gradients). The following acquisition parameters were employed: 13.05 µs 90-degree pulse, 5 s delay, 64 K data points, 64 scans, 3.89 s acquisition time, and 10.03 ppm spectral width.

The processing of 1D ^1^H NMR spectra was carried out using the Chenomx NMR Suite Professional 7.7 software (Chenomx Inc., Edmonton, AB, Canada). This involved phasing baseline correction and pH calibration using imidazole resonances. Spectra were referenced to the DSS methyl peak at 0.00 ppm, which also served as a chemical shape indicator and internal standard for quantitation.

#### 2.7.3. Metabolite Identification and Quantitation

The procedure was carried out based on the methodology described by Antonelo et al. [36] and Cônsolo et al. [37]. Metabolites in the 1D ^1^H-NMR spectra were identified using Chenomx NMR Suite Professional 7.7. software (Chenomx Inc., Edmonton, AB, Canada) with a built-in 1D spectral library. 43 metabolites were quantified in the spectra of meat extracts using the profiler module. Quantitation was performed by comparing the area of selected metabolite peaks with the area under the DSS methyl peak, corresponding to a known concentration of 0.5 mM in each sample. Metabolites were individually identified using Chenomx NMR Suite Professional 7.7. software with a built-in 1D spectral library. The resulting metabolite concentrations were exported to Excel as a table, and sample identifiers were subsequently added.

#### 2.7.4. Statistics of Metabolomic Analysis

Metabolomic data were analyzed using R version 4.4.0 software (R Found., Vienna, Austria) and Metabo Analyst 5.0. The metabolite concentration table was uploaded to Metabo Analyst, and the data were log-transformed and Pareto-scaled before analysis. To rank metabolites based on their importance in discriminating groups (control vs. supplemented animals and control vs. animals fed 75 mg/kg DM), the variable importance in the projection (VIP) was used in the PLS-DA model. Metabolites with the highest VIP values are the most powerful group discriminators. Typically, VIP values > 1 are considered significant, and VIP values > 2 are highly significant. 

The metabolites data set was used to construct an overview of enriched metabolite sets, indicating the most important pathways between treatments. The compound names were standardized according to the Kyoto Encyclopedia of Genes and Genomes ID. Given the exploratory nature of this study, we included pathways with a raw *p* value of <0.1 to identify them as having high impact and interest.

### 2.8. Statistical Analyzes

The data on intake, performance, carcass attributes, and meat quality were analyzed using SAS (Version 9.4, SAS Institute, Cary, NC 2015, USA). The normality of the residues and homogeneity of variances were verified using the PROC UNIVARIATE.

The data were analyzed by PROC MIXED according to the following model:Yij = µ + Ai + Sj + eij
where: 

Yij = dependent variable, µ = general average, Ai = random effect of animal (I = 1 to 48); Sj = additive fixed effect (j = 1 to 2), and e_ij_ = experimental error. 

The data of ingestive behavior were analyzed using SAS (Version 9.4, SAS Institute, Cary, NC 2015, USA). The normality of residues and homogeneity of variances were verified using PROC UNIVARIATE.

The data were analyzed by PROC MIXED according to the following model:Y_ijk_ = µ + Ai + Sj + T_k_ + S_j_*T_k_ + e_ijk_
where: 

Yij = dependent variable, µ = general average, Ai = random effect of animal (I = 1 to 48); Sj = additive fixed effect (j = 1 to 2); Tk = time random effect (k = 1 a 9); S_j_*T_k =_ interaction effect of additive and time, and e_ijk_ = experimental error.

The degrees of freedom were corrected by DDFM = kr. The data obtained were subjected to analysis of variance using the PROC MIXED command of SAS, version 9.4 (SAS, 2015), adopting a significance level of 5%, i.e., *p* < 0.05. The means were analyzed by orthogonal contrasts, where C1 (control vs. 3-nitrooxypropanol) and C2 (75 vs. 100 mg/kg DM of 3-nitrooxypropanol).

## 3. Results

### 3.1. Intake and Animal Performance 

Animals that received 3-NOP exhibited higher final body weight (BW) (*p* = 0.046) and ADG (*p* = 0.038) in comparison to animals in the control group. Specifically, steers supplemented with BV75 demonstrated higher BW (*p* = 0.036) and ADG (*p* = 0.025) in comparison to those supplemented with BV100 (Table 2). No significant differences were observed in DMI (kg/day and %BW) between the supplemented animals in comparison to the control group. However, the BV75 group exhibited a higher DMI (kg/day and %BW) (*p* = 0.047) compared to the BV100 group. The addition of 3-NOP resulted in increased hot carcass weight (HCW) (*p* = 0.037), ADGcarcass (*p* = 0.038), and carcass conversion (*p* = 0.023). Additionally, steers supplemented with BV75 showed greater average carcass gain (ACGcarcass) (*p* = 0.023).

### 3.2. Ingestive Behavior of Animals 

The experimental treatments did not exert a significant influence (*p* ≥ 0.651) on the ingestive behavior of the evaluated animals. However, a notable effect of the experimental period (*p* < 0.0001) was observed for intake time (min/day), visits to the trough (n), and dry matter intake time (min/kg and kg/trough visits). Additionally, an interaction (*p* = 0.034) was observed between the treatment and experimental periods for visits to the trough (n) (Table 3).

Steers supplemented with BV75 exhibited a significant period effect prolonged trough occupancy (min/day) starting from the 40th day of the experimental period, and this behavior persisted until the 80th day of the evaluation period (Figure 1). Conversely, animals in the BV100 group did not show any differences in comparison to the control group throughout the entire experimental period. Overall, all experimental groups demonstrated an increase in the time spent in the trough from the 30th day of the experimental period.

The BV75 treatment exhibited an effect on the period with the highest frequency of visits to the trough starting from the 50th day of the experimental period, and this behavior persisted until day 86 of the evaluation period (Figure 2), as well as an interaction between the variables analyzed, when compared with the other treatments. Generally, all experimental groups exhibited an effect for the period, resulting in a reduction in the number of visits to the trough from the 40th day of the experimental period.

Regarding dry matter intake (min/kg and kg/trough visits), no differences were observed between the experimental groups throughout the experimental period (Figure 3 and Figure 4). However, an increase in dry matter intake (kg/trough visits) was noted throughout the experimental period.

### 3.3. Gas Emission

Supplementation with 3-NOP resulted in lower methane emissions (g/day) (*p* < 0.0001) and increased hydrogen emissions (g/day) (*p* < 0.0001) compared to the control group. Supplemented steers with 3-NOP emitted an average of 38.2% less methane than non-supplemented animals, and hydrogen emissions (g/day) were 4.45 times greater for steers supplemented with 3-NOP. Beef cattle receiving 3-NOP exhibited reduced methane emissions (*p* < 0.0001) per kg of dry matter intake (CH_4_:IMS), average daily gain (CH_4_: ADG), average carcass gain (CH_4_:ADGcarcass), and carcass conversion (CH_4_:carcass conversion) compared to the control group. When comparing the emission values between treatments with 3-NOP, animals supplemented with BV100 demonstrated lower methane emissions per average daily gain (*p* < 0.0001) compared to BV75 (Table 4).

### 3.4. Quality and Chemical Composition of the Meat

The addition of 3-NOP resulted in lower pH (*p* = 0.002), higher water retention capacity (*p* = 0.041), lower shear force (*p* = 0.042), and higher lipid content (*p* = 0.031) in meat compared to animals in the control group (Table 5). No differences were observed between the experimental groups in meat color (a*), fat color (b*), and luminosity (L*).

The use of 3-NOP in the steers’ diet influenced the values of ether extract (EE), which were higher in treatments that received 3-NOP (*p* = 0.031), with no significant difference observed between the experimental groups that were supplemented.

Steers supplemented with 3-NOP presented a higher concentration of some specific fatty acids in meat, exhibiting higher concentrations of C14:1 (myristoleic), C16:1 (palmitoleic), C17:1 (heptadecenoic), C18:2 CLA (cis-9, trans-11, rumenic), C18:3 ω3 (linolenic), and C20:1 (eicosenoic) in comparison to non-supplemented animals (Table 6). No significant differences were observed between the groups of animals concerning the sum, enzymatic activity, indices, and relationships between the fatty acids presented in Table 6. Table 7 did not present statistical differences in the sum, relationships, enzymatic activity, and fatty acid profile index of meat from Nelore animals supplemented with 3-Nitrooxypropanol in confinement diets.

### 3.5. Muscle Metabolomics 

The sPLSDA analysis was conducted to enhance the visualization of differences in group metabolism. The partial separation observed among groups indicated that 3-NOP induced alterations in meat metabolism (Figure 5). Specifically, these differences are evident in the VIP scores when comparing the control group with the 3-NOP group. In this comparison, metabolites such as 3-hydroxybutyrate, citrate, choline, creatine, L-acetylcarnitine, methanol, butyrate, aspartate, cis-aconitate, and tyrosine exhibited VIP scores > 1 (Figure 6A). Notably, among these metabolites, only 3-hydroxybutyrate demonstrated a higher concentration in the CON group. Similarly, when comparing the control group with animals fed 75 mg/kg DM, metabolites including 3-hydroxybutyrte, pyruvate, tyrosine, glutamate, dimethylamine, acetate, aspartate, citrate, and choline showed VIP scores > 1 (Figure 6B). The majority of these compounds displayed higher concentrations in the meat of animals on the control diet, with the exception of dimethylamine, aspartate, citrate, and choline (Figure 7).

Irrespective of the specific comparisons made, the overall metabolite profile implies a more favorable energetic status in meat from treated animals. This improvement is attributed to the enhanced efficiency of rumen metabolism achieved by feeding the animals 3-NOP. It is noteworthy that the use of this product resulted in an average reduction of 38.2% in methane emissions.

## 4. Discussion

Nellore cattle, which are fed diets supplemented with 3-NOP, showed a reduction in enteric CH_4_ emissions. This confirms the hypothesis that 3-NOP acts as an inhibitor of methanogenesis [14]. The potential of 3-NOP has been confirmed in previous studies in beef cattle [11,14,16,17,39] and others [3,5,10]. Therefore, the present study aimed to investigate the use of 3-nitrooxypropanol (3-NOP) in the diet of confined cattle in Brazil, with a focus on finding a dose at which the tested additive achieves a balance between methane reduction and animal performance. This indicates a level that does not negatively influence production and is viable for commercial feedlots. The reduction in CH_4_ production is directly linked to the dose supplied to animals [18], and numerous studies have explored various doses of 3-NOP, yielding different results. It is important to note that the dose can influence dry matter intake (DMI) and, consequently, animal performance. In our study, animals supplemented with 3-NOP emitted approximately 38.20% less methane than non-supplemented animals.

The BV100 group supplemented with 100 mg/kg DM showed the lowest emission, with a 42.71% reduction when compared to the control, whereas the BV75 group supplemented with 75 mg/kg DM showed a 33.7% reduction compared to the control group. Interestingly, Vyas et al. [39] reported that a dose of 75 mg/kg DM of 3-NOP showed no significant difference in CH_4_ emissions compared to the control. The efficiency of 3-NOP as an inhibitor is attributed to its mechanism of action, which involves temporary inactivation of the MCR active site, thereby inhibiting methanogenesis [2,8,40]. In other words, this compound operates at the final stage of methanogenesis, preventing the formation of CH_4_ [41]; however, inhibiting CH_4_ formation in the rumen environment may result in an increased concentration of H_2_. This is due to the fact that methanogenesis serves as the primary sink for H_2_ in the rumen [9], leading to an accumulation of H_2_ [42], nonetheless, this excess H_2_ can be directed toward alternative fermentation pathways. Consequently, the quantity of expelled H_2_ by the animals increases, as evidenced in the current study, where hydrogen emission was 4.45 greater for steers supplemented with 3-NOP. Gruninger et al. [2] observed a 37-fold increase in H_2_ in cattle supplemented with 3-NOP compared to the control group.

Although H_2_ is not entirely reallocated and accumulates in the ruminal environment, increasing emissions, the theory suggests that a portion of this H_2_ is redirected to alternative fermentation pathways that provide greater energetic benefits to animals, thereby improving overall performance [43]. It is noteworthy that despite the accumulation of H_2_ due to the addition of 3-NOP, nutrient digestibility in the digestive tract of ruminants remains unaffected [9]. A study conducted by Araújo et al. [14], involving 138 confined animals administered two doses of 3-NOP, with one group receiving 100 mg/kg DM and the second group receiving 150 mg/kg DM of 3-NOP, observed a consistent reduction of 49.3% (g/d) in CH_4_ emissions during the assessment periods, irrespective of the 3-NOP dosage analyzed. Alemu et al. [17] progressively adapted the animals to 3-NOP doses, with a final concentration of 100 mg of 3-NOP/kg DM for 7 to 10 days, 150 mg of 3-NOP/kg DM for 7 days and a final dose of 200 mg/kg DM until the conclusion of the study, the methane production (g/d) exhibited reductions of 17.4%, 28.8% and 28.1%, respectively, compared to the control, providing backgrounding diets. The results found by Almeida et al. [44] show that the CH_4_ yield showed a quadratic decrease as the 3-NOP inclusion rate increased, resulting in 65.5%, 80.2%, 85.3%, and 87.6% reduction in CH_4_ production compared to the control, for 50, 75, 100, and 125 mg of 3-NOP/kg DM, respectively. Concurrent with the decrease in CH_4_, there was an increase in H_2_ emissions; however, due to the study design, comparing responses across doses is challenging. Some studies have reported greater methane reduction at certain tested dosages [39].

Romero-Pérez et al. [15] observed a reduction in the total number of methanogens, consistent with the decrease in CH_4_ emissions. In Lopes et al. [45], the inhibitor showed no significant effect on the rumen composition of archaea; however, there was a tendency to reduce the proportion of methanogenic cell counts in the entire rumen content. Consequently, there are indications that besides decreasing methanogenic activity, inhibition of methanogenesis can also result in a decrease in the methanogenic population [42]. It is evident that 3-NOP does not compromise feed intake, productive performance, or product quality in ruminants [9]. Even when DMI decreases or tends to decrease with increasing 3-NOP supplementation levels, there are no adverse effects on animal performance [13]. In the current study, DMI did not show a significant difference between animals in control and supplemented groups [46]; however, within the groups that received different doses of 3-NOP, there was a noticeable drop in DMI in the steers in the BV100 group. While several factors can influence intake, the reduction in intake coincides with higher levels of 3-NOP supplementation in beef cattle [13].

Some studies have indicated that an increased addition of 3-NOP can influence DMI [11]. High doses (200 mg/kg DM) were shown to reduce DMI during the backgrounding phase and tended to decrease intake during the finishing phase compared to low doses (100 mg/mg DM) [16]. The results obtained in the DMI align with the digestive behavior graphs of the experimental groups. Despite showing no difference between treatments, there was an effect on time, indicating that the BV75 animals had greater intake over the intake time, trough visits, and DMI compared to the BV100 group. However, the reduction in DMI itself is not a concern if it results in the same live weight gain in the animal, suggesting an improvement in feed use efficiency [47].

The DMI data obtained in this study contrasts with the findings of Vyas et al. [39], who reported no effects on the DMI of steers supplemented with levels ranging from 0 to 200 mg/kg DM. Feeding with 3-NOP, despite showing a tendency to reduce DMI, demonstrated an improvement in the G:F ratio by 2.5% [17]. In a study involving 138 confined animals and providing two doses of 3-NOP (100 mg/kg DM and 150 mg/kg DM), Araújo et al. [14] found no negative effects on animal performance. According to Jayanegara et al. [9], the improved G:F ratio observed in beef cattle may imply a more efficient use of energy by animals, possibly through the reduction of energy loss via CH_4_ emissions. This could explain the study’s data where, despite lower DMI values compared to the control group, the animals in the BV75 group performed better, exhibiting a higher final weight, ADG, and ADGcarcasss. In contrast, the animals in the BV100 group were unable to achieve the same level of efficiency despite a greater reduction in CH_4_.

The significant reduction in GE lost as CH_4_ by 42.5% with 3-NOP is noteworthy [14]. This decrease in the intake of GE lost as CH_4_ is associated with other fermentation pathways that also utilize CO_2_ and hydrogen as substrates, directing them toward alternative metabolic pathways [11]. These pathways are considered more energy-efficient because methane production represents a loss of energy, thereby enhancing the overall efficiency [11,12,13,48].

Steers supplemented with 3-NOP exhibited a greater hot carcass weight (HCW) compared to the control group, accompanied by a lower subcutaneous fat (SF) value. These characteristics are closely linked to the quality of beef meat and have a notable impact on sensory attributes such as tenderness and juiciness. Tenderness, a crucial phenotypic characteristic of beef quality, is influenced by various factors throughout the animal’s life, including breed, muscle tissue, and environmental conditions, which affect the antemortem, rigor mortis, and post-mortem periods [49].

The lipid composition of beef, similar to that of milk fat, mirrors the ruminal metabolism of dietary lipids [50,51]. Nellore meat is characterized by attributes beneficial to human health, such as a high profile of bovine fatty acids in the n-3 series, along with an appropriate dietary balance based on n-6/n-3 proteins and PUFA/SFA levels [52]. Although carcass characteristics did not appear to be negatively influenced, further studies are necessary to fully elucidate the impact of 3-NOP on ruminal fermentation and its consequent influence on the chemical composition, lipid composition, and overall quality of beef. The VIP score has proven to be a valuable tool for understanding variations in metabolite concentrations, offering insights into the distinctions between the meat of non-supplemented and supplemented cattle. Figure 6 depicts significantly elevated levels of 3-hydroxybutyrate and pyruvate in the meat of non-supplemented cattle compared to the supplemented group.

The 3-hydroxybutyrate and pyruvate are closely associated with energy metabolism. Notably, 3-hydroxybutyrate and pyruvate, recognized as essential energy sources, play a critical role in carbohydrate metabolism in cattle and exhibit a strong correlation with growth rate and overall animal performance [53]. Specifically, 3-hydroxybutyrate, acknowledged as a pivotal biomarker of animal performance in fat tissue metabolism, serves as an alternative energy source in the absence of sufficient blood glucose, particularly during periods of starvation or illness. Its multifaceted role extends to providing acetoacetyl-CoA and acetyl-CoA for cholesterol, fatty acids, and complex lipid synthesis.

Studies by Hammon et al. [54] have linked higher concentrations of 3-hydroxybutyrate to genetic lines with lighter carcasses and lower body fat proportions, emphasizing its relevance in specific cattle breeds. Additionally, Yanibada et al. [55] observed decreased levels of 3-hydroxybutyric acid in cows supplemented with 3-nitrooxypropanol, which is associated with a better energy balance. The observed variations in these metabolites shed light on the metabolic shifts induced by 3-NOP supplementation, providing valuable insights into the potential impact on the energy metabolism and performance of cattle.

Pyruvate, an end product of glycolysis, resides at the intersection of multiple pathways, including glycolysis, gluconeogenesis, and the tricarboxylic acid (TCA) cycle [56]. The lower pyruvate levels in the meat of animals fed 3-NOP may indicate a greater demand for energy to enhance muscle mass and, consequently, greater weight gain. The treated animals in the present study exhibited a 6% higher average daily gain and a 6 kg greater hot carcass weight, showing that 75 mg/kg of 3-NOP altered meat metabolism.

In muscle cells, pyruvate undergoes several biochemical pathways depending on oxygen availability. Under anaerobic conditions, pyruvate is converted to lactate by lactate dehydrogenase, regenerating the NAD+ needed for glycolysis. This process, known as anaerobic glycolysis, allows ATP production to continue despite the low oxygen levels. Under aerobic conditions, pyruvate enters the mitochondria and is converted to acetyl-CoA by pyruvate dehydrogenase. Acetyl-CoA then enters the citric acid cycle (Krebs cycle) to produce NADH and FADH2, which donate electrons to the electron transport chain, leading to oxidative phosphorylation and efficient ATP production. These pathways ensure that muscle cells can generate energy under varying conditions [57].

The observed decrease in metabolites associated with energy metabolism in animals supplemented with 3-NOP suggests a redirection of energy resources, potentially conserved from reduced methane emissions, toward crucial processes such as growth and overall performance. This dynamic reallocation signifies a more efficient use of available energy, in contrast with the wasteful release of energy in the form of methane. Consequently, supplemented cattle may utilize this redirected energy for enhanced growth and performance, aligning with sustainable farming practices and emphasizing the economic and ecological benefits of additive methane reduction. In essence, these detailed metabolic insights significantly contribute to a comprehensive understanding of how methane reduction supplementation influences meat composition and plays a pivotal role in optimizing animal efficiency. These findings have substantial implications for livestock management, highlighting the potential for improved animal performance and reduced environmental impacts through targeted interventions in metabolic pathways related to methane emissions in cattle.

A noteworthy observation from this study is the elevated choline concentration in animals supplemented with 3-NOP. Choline, a vitamin-like compound, functions primarily as a phospholipid and plays a crucial role in cell membrane integrity, lipid digestion, and transport [58]. However, the mechanisms by which choline improves cattle growth are not fully understood; they are possibly related to its role in lipid mobilization and transport [59], which, in turn, can be correlated with lipolysis for energy generation, supporting greater muscle mass for animals fed 75 mg/kg DM. These results suggest an association with cattle growth rate, which is in agreement with other studies [53,58,59]. These insights deepen our understanding of how rumen modulation shapes meat metabolites and contribute to a more comprehensive understanding of system-wide metabolism, impacting animal efficiency. Creatine was also an important metabolite for discriminating between groups, being observed in different amounts in animals treated with 3-NOP and in untreated animals. It is an organic compound that plays a relevant function in cellular energy metabolism, supplying high-energy phosphate groups to the cell through the creatine kinase–phosphocreatine system. Additionally, creatine is well known as a biomarker for the total amount of muscle mass [60], which aligns with the data from the present study. At least, the animals fed 75 mg/kg DM presented greater average daily gain and hot carcass weight compared to the control group, as previously explained. Despite the focus on animal performance, metabolomic meat data can also be utilized to speculate on meat quality parameters. The increased concentration of citric acid in supplemented animals is noteworthy, as citric acid has the dual benefit of improving the water-holding capacity and tenderness of beef muscle while inhibiting lipid oxidation [61]. This may partly explain why the meat from the supplemented animals in the present study was more tender and had a higher water capacity.

Aspartic acid is also a metabolite correlated with meat quality, as it has been associated with flavor sensation and umami. Its higher concentration in the meat of animals fed 3-NOP could suggest better overall acceptability for consumers. However, despite these observations, there is still a lack of knowledge about how rumen modulation to decrease methane emission can alter quality parameters at a molecular level. Additionally, further studies with sensory panels need to be conducted to confirm these findings.

The metabolites were enriched in metabolomic pathways, and the most significant pathway was pyruvaldehyde degradation (*p* = 0.072), identified for the differentiation between animals fed the control diet and those fed 75 mg/kg DM (Figure 7). Pyruvaldehyde is an intermediate of L-ascorbic acid (ASA) degradation, and it may react with cysteine or its degradation products to generate a variety of aroma compounds during the Maillard reaction [62].

Finally, based on the results obtained, we conclude that 3-nitrooxypropanol (3-NOP), Bovaer^®^, is a compound capable of reducing emissions, contributing to more sustainable and environmentally friendly livestock farming. However, more studies will be needed to compare meat quality and metabolomic data. In addition, research with a larger number of animals is needed to increase the precision. Therefore, given its potential to reduce methane production and improve animal performance based on this trial outcome, the dosage of 75 mg/kg DM of 3-NOP seems to be better for use in finishing diets.

## 5. Conclusions

Feeding 3-nitrooxypropanol resulted in a reduction in methane emissions, regardless of the dose used. Therefore, we conclude that these dosages are effective for use in commercial feedlots, as they simultaneously reduce emissions and improve animal performance. Importantly, these positive effects were achieved without compromising the meat quality and characteristics of commercial interest in feedlot Nellore animals.

## Figures and Tables

**Figure 1 animals-14-02576-f001:**
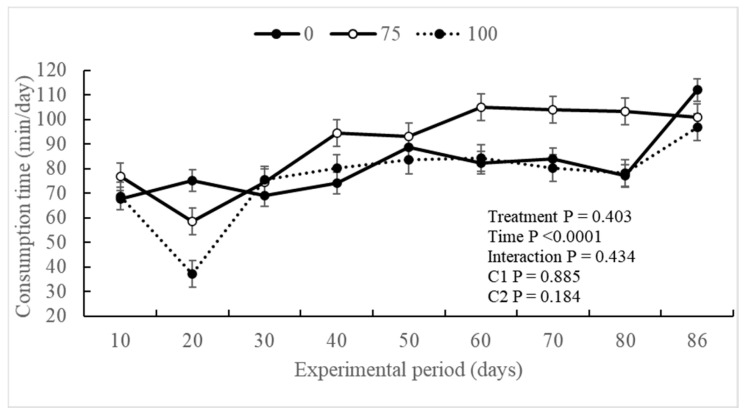
Intake time (min/kg DM) according to the experimental treatments throughout the experimental period of Nellore animals supplemented with 3-Nitrooxypropanol in feedlot diets.

**Figure 2 animals-14-02576-f002:**
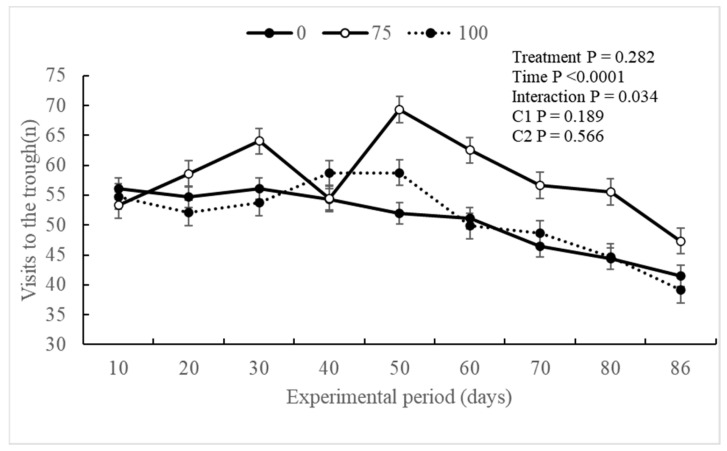
Visits to the trough (n) according to the experimental treatments throughout the experimental period of Nellore animals supplemented with 3-Nitrooxypropanol in feedlot diets.

**Figure 3 animals-14-02576-f003:**
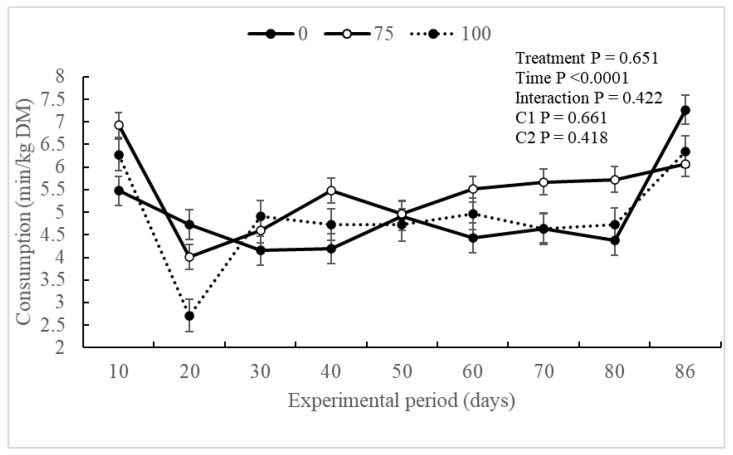
Dry matter intake (min/kg) according to the experimental treatments throughout the experimental period of Nellore animals supplemented with 3-Nitrooxypropanol in feedlot diets.

**Figure 4 animals-14-02576-f004:**
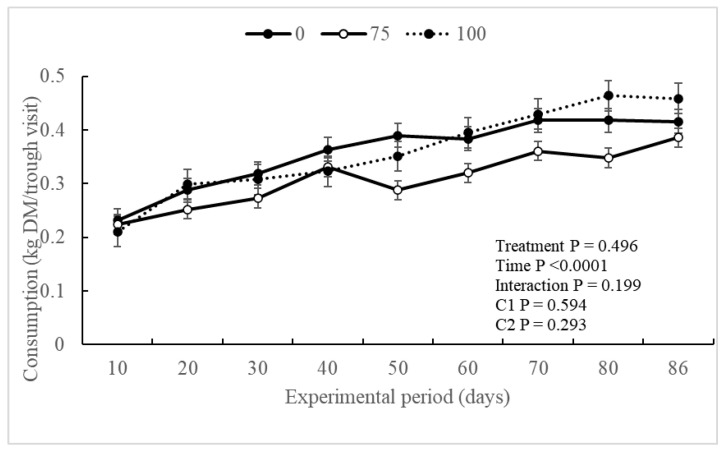
Dry matter intake (kg/trough visit) according to experimental treatments throughout the experimental period.

**Figure 5 animals-14-02576-f005:**
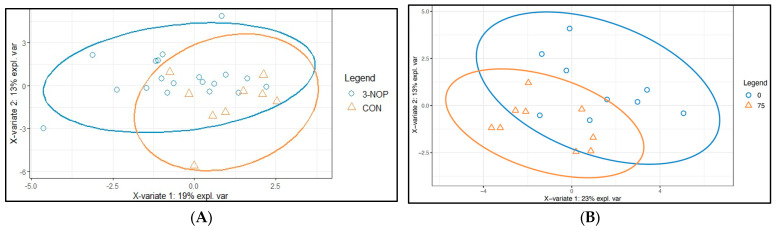
sPLSDA score plot of meat metabolomics profile from animals. (**A**) Control diets (CON) vs. animals fed with 3-NOP regardless of the concentration; and (**B**) Control diet vs. animals fed 75 mg/kg MS of 3-NOP.

**Figure 6 animals-14-02576-f006:**
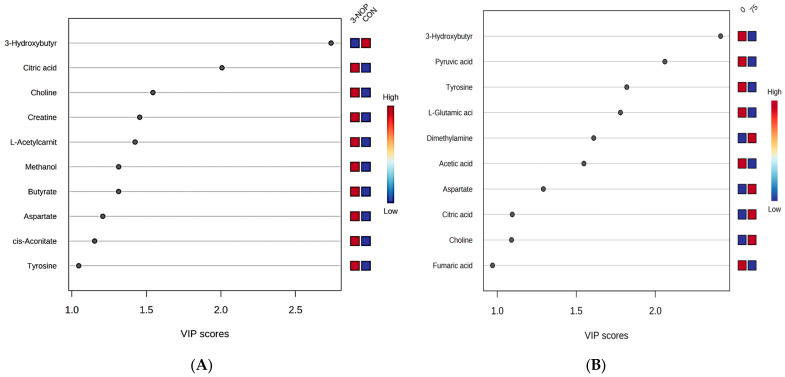
Variation important projection of metabolites ranked according to metabolite importance in group differentiation. The dots in the figure represent the VIP score for each metabolite evaluated. (**A**) Control diets (CON) vs. animals fed with 3-NOP; and (**B**) Control diet vs. animals fed 75 mg/kg MS of 3-NOP.

**Figure 7 animals-14-02576-f007:**
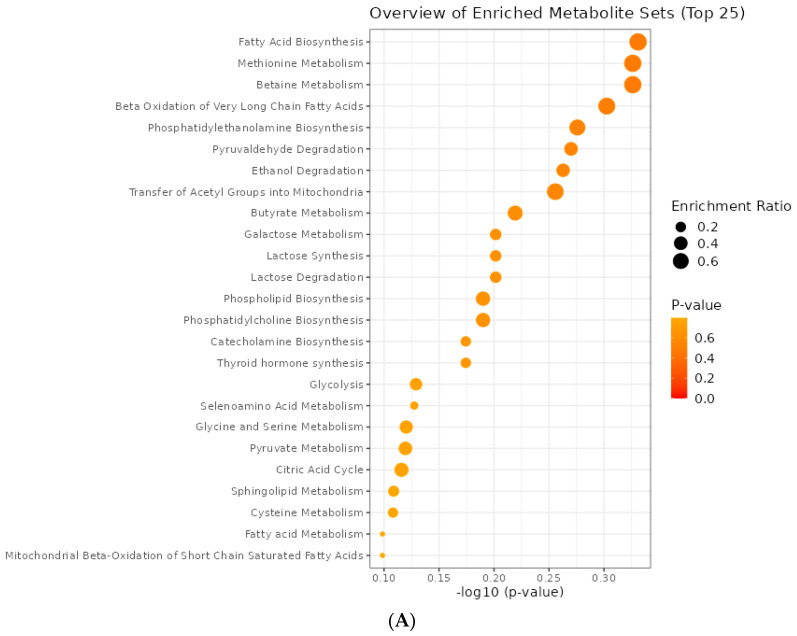
Overview of enriched metabolite sets between animals fed control and 3-NOP (**A**) and animals fed control and 75 mg/kg MS of 3-NOP (**B**).

**Table 1 animals-14-02576-t001:** Ingredients and nutritional composition of the experimental diets.

Ingredient (%)	Diets		
	Adaptation I (0–7 d)	Adaptation II (7–14 d)	Finishing
Sugarcane bagasse	17.84	13.57	9.60
Ground corn	46.22	36.77	39.26
Wet corn grain silage	0.00	15.00	18.00
Cotton pie	21.50	21.42	22.00
Cottonseed meal	11.50	10.30	8.00
Mineral and vitamin supplements ^1,2^	2.95	2.94	2.94
Protected urea	0.00	0.00	0.10
Urea	0.00	0.00	0.10
Chemical composition (g/kg DM)			
Dry matter	559.05	603.80	619.47
Organic matter	934.92	936.69	936.96
Crude protein	141.46	141.18	143.36
Neutral detergent fiber	352.98	314.98	282.67
Acid detergent fiber	145.90	131.36	119.96
Lignin	54.38	48.48	43.69
Starch	308.34	339.90	373.10
Ethereal extract	42.76	45.09	47.21
Ash	65.08	63.31	63.04
Non-fibrous carbohydrates	397.71	435.44	463.72
Total digestible nutrientes	622.49	634.93	644.57
Net energy gain (Mcal/kg DM)	2.35	2.41	2.45

^1^ Mineral and vitamin supplement containing dietary treatments: MON = sodium monensin (26 mg/kg DM); BEO = blend of essential oils (90 mg/kg DM); Sodium monensin (Rumensin) was obtained from Elanco Animal Health, Indianapolis, IN. A blend of essential oils (CRINA Ruminants) was provided by DSM-Firmenich, Basel, Switzerland. ^2^ Mineral and vitamin supplement was composed (DM basis) of 262 g/kg Ca, 18 g/kg P, 23 g/kg S, 17 g/kg Mg, 70 g/kg Na, 20 mg/kg Co, 455 mg/kg Cu, 14 mg/kg Cr, 38 mg/kg I, 1269.99 mg/kg Mn, 14 mg/kg Se, 1700 mg/kg Zn, 83,400.07 IU/kg vitamin A, 16,679.98 IU/kg vitamin D3, 170 IU/kg vitamin E. Manufactured by DSM-Firmenich, São Paulo, Brazil.

**Table 2 animals-14-02576-t002:** Productive performance and carcass characteristics of Nellore animals supplemented with 3-Nitrooxypropanol in feedlot diets.

Item	Treatments ^1^			SEM ^2^	*p* ^3^ Value	
	CON	BV-75	BV-100		C1	C2
BW_Initial_, kg	406.30	409.60	405.89	5.719	-	-
BW_Final_, kg	546.70	558.30	543.22	7.116	0.046	0.036
ADG ^4^, kg day	1.63	1.73	1.60	0.045	0.038	0.025
DMI ^5^, kg/day	11.32	11.14	10.61	0.208	0.323	0.028
DMI/BW ^6^, %BW	2.38	2.31	2.24	0.038	0.204	0.047
DMI:ADG	7.05	6.52	6.71	0.156	0.442	0.612
G:F	0.15	0.15	0.15	0.003	0.339	0.657
HCW_initial_ ^7^, kg	197.70	199.47	197.81	3.029	0.346	0.623
HCW_final_ ^8^, kg	306.65	312.30	305.33	4.029	0.037	0.564
CY ^9^, %	56.08	55.97	56.21	0.172	0.979	0.587
ADG ^10^, kg of carcass	1.27	1.31	1.26	0.027	0.038	0.023
Carcass conversion, kg of DM/@ produced	135.27	128.04	127.81	2.544	0.023	0.547

^1^ Inclusion of BOVAER^®^ (3-nitrooxypropanol) (mg/kg DM),. ^2^ SEM (standard error of the mean). ^3^ C1 (control vs. BOVAER^®^); C2 (75 vs. 100 mg/kg DM BOVAER^®^). ^4^ Average daily gain. ^5^ Dry matter intake. ^6^ Matter intake by live weight (%BW). ^7^ Initial hot carcass weight. ^8^ Final hot carcass weight. ^9^ Carcass yield. ^10^ Average carcass gain.

**Table 3 animals-14-02576-t003:** Ingestive behavior of Nellore animals supplemented with 3-Nitrooxypropanol in feedlot diets.

Item	Treatments ^1^			SEM ^2^	*p* Value ^3^				
	CON	BV-75	BV-100		Trat	Time	INT	C1	C2
Intake time (min/day)	81.75	90.05	76.05	2.329	0.403	<0.0001	0.434	0.885	0.184
Visits to trough (n)	50.73	57.98	51.13	1.041	0.282	<0.0001	0.034	0.189	0.566
Dry Matter Intake									
Minutes/kg	4.90	5.44	4.89	0.144	0.651	<0.0001	0.422	0.661	0.418
Kg/trough visit	0.358	0.309	0.362	0.008	0.496	<0.0001	0.199	0.594	0.293

^1^ Inclusion of BOVAER^®^ (3-nitrooxypropanol, mg/kg DM). ^2^ SEM (standard error of the mean). ^3^ Treatment effect; time and interaction Treatment*Time; C1 (control vs. BOVAER^®^); C2 (75 vs. 100 mg/kg DM BOVAER^®^).

**Table 4 animals-14-02576-t004:** Methane emission of Nellore animals supplemented with 3-Nitrooxypropanol in feedlot diets.

Item	Treatments ^1^			SEM ^2^	Value of *p* ^3^	
	0	75	100		C1	C2
CH_4,_ g/day	204.45	135.56	117.14	7.859	<0.0001	0.041
H_2,_ g/day	1.01	4.27	4.73	0.083	<0.0001	0.390
CH_4_:DMI	18.21	12.24	11.08	0.681	<0.0001	0.190
CH_4_:ADG	129.04	80.25	73.64	5.916	<0.0001	<0.0001
CH_4_:ADG_carcass_	164.14	104.71	93.93	7.017	<0.0001	0.287
CH_4_:Carcass conversion	1.53	1.06	0.93	0.058	<0.0001	0.107

^1^ Inclusion of BOVAER^®^ (3-nitrooxypropanol) (mg/kg DM). ^2^ SEM (standard error of the mean). ^3^ C1 (control vs. BOVAER^®^); C2 (75 vs. 100 mg/kg MS BOVAER^®^).

**Table 5 animals-14-02576-t005:** Quality and chemical composition of meat of Nellore animals supplemented with 3-Nitrooxypropanol in feedlot diets.

Item	Treatments ^1^			SEM ^2^	Value of *p* ^3^	
	0	75	100		C1	C2
pH	5.68	5.62	5.61	0.015	0.002	0.421
Water retention capacity, %	76.62	78.22	79.91	0.794	0.041	0.381
Cooking losses, %	41.98	41.34	40.44	0.654	0.125	0.458
Shear force, kg/cm^2^	7.66	7.23	7.00	0.015	0.042	0.496
Color						
a*	14.90	14.99	14.97	0.201	0.235	0.125
b*	3.27	3.66	3.40	0.159	0.458	0.526
L*	35.53	35.36	35.25	0.194	0.601	0.825
Chemical composition						
Moisture, %	70.90	70.82	71.13	0.097	0.204	0.742
Ethereal extract, %	3.85	5.51	5.73	0.284	0.031	0.235

^1^ Inclusion of BOVAER^®^ (3-nitrooxypropanol mg/kg DM). ^2^ SEM (standard error of the mean). ^3^ C1 (control vs. BOVAER^®^); C2 (75 vs. 100 mg/kg MS BOVAER^®^).

**Table 6 animals-14-02576-t006:** Individual meat fatty acid profiles of Nellore animals supplemented with 3-Nitrooxypropanol in feedlot diets.

Item	Treatments ^1^			SEM ^2^	*p* Value ^3^	
	0	75	100		C1	C2
Fatty Acids (g/100 g)						
C10:0	0.109	0.109	0.111	0.001	0.421	0.168
C12:0	0.210	0.205	0.216	0.003	0.945	0.200
C14:0	2.423	2.459	2.416	0.022	0.766	0.447
C14:1	0.361	0.372	0.381	0.004	0.033	0.390
C15:0	0.104	0.106	0.104	0.001	0.621	0.394
C16:0	21.69	21.77	21.69	0.038	0.626	0.460
C16:1	1.63	1.65	1.68	0.011	0.037	0.236
C17:0	1.36	1.34	1.34	0.011	0.324	0.945
C17:1	0.834	0.879	0.857	0.007	0.026	0.198
C18:0	17.45	17.42	17.45	0.042	0.872	0.725
C18:1	48.08	48.05	48.02	0.060	0.736	0.835
C18:2 ω6	3.59	3.57	3.60	0.023	0.992	0.533
C18:2 CLA	0.274	0.286	0.291	0.003	0.036	0.552
C18:3 ω3	0.144	0.146	0.149	0.002	0.019	0.631
C20:0	0.143	0.141	0.142	0.002	0.783	0.873
C20:1	0.105	0.109	0.110	0.001	0.035	0.672
C20:2	0.114	0.110	0.112	0.001	0.094	0.255
C20:3 ω3	1.37	1.33	1.34	0.011	0.175	0.884
C20:3 ω6	0.112	0.110	0.112	0.001	0.459	0.204
C20:4	0.145	0.141	0.140	0.003	0.602	0.920
C20:5 ω3	0.106	0.104	0.102	0.001	0.123	0.366
C22:1	0.347	0.352	0.353	0.001	0.294	0.994

^1^ Inclusion of BOVAER^®^ (3-nitrooxypropanol mg/kg DM). ^2^ SEM (standard error of the mean). ^3^ C1 (control vs. BOVAER^®^); C2 (75 vs. 100 mg/kg DM BOVAER^®^).

**Table 7 animals-14-02576-t007:** Sum, relationships, enzymatic activity, and meat fatty acid profile index of Nellore animals supplemented with 3-Nitrooxypropanol in feedlot diets.

Item	Treatments ^1^			SEM ^2^	*p* Value ^3^	
	0	75	100		C1	C2
Ʃ 10-C a 14-C ^4^	3.10	3.14	3.12	0.022	0.530	0.716
Ʃ acima de 16-C ^5^	97.62	97.63	97.64	0.024	0.985	0.948
Ʃ AGS ^6^	43.50	43.55	43.49	0.047	0.874	0.602
Ʃ AGI ^7^	57.22	57.23	57.26	0.048	0.879	0.755
Ʃ AGMI ^8^	51.35	51.42	51.40	0.061	0.682	0.939
Ʃ AGPI ^9^	5.86	5.80	5.85	0.034	0.601	0.572
Ʃ AGCI ^10^	2.30	2.32	2.30	0.013	0.735	0.520
Ʃ AG ω3^11^	0.362	0.360	0.363	0.002	0.935	0.675
Ʃ AG ω6 ^12^	4.96	4.90	4.94	0.003	0.627	0.607
Ratio sat/insat ^13^	1.31	1.32	1.32	0.001	0.932	0.658
Ratio sat/insat 18-C ^14^	0.336	0.334	0.335	0.110	0.514	0.706
Ratio ω6: ω3 ^15^	13.73	13.64	13.63	0.022	0.703	0.991
Product/substrate relationship ^16^						
C:14:1/14:0	6.73	6.63	6.37	0.102	0.280	0.301
C:16:1/16:0	13.32	13.16	12.86	0.095	0.136	0.196
C: 18:1/18:0	0.363	0.362	0.363	0.001	0.867	0.772
Δ ^9^ desaturase C16:0	6.99	7.06	7.21	0.047	0.125	0.196
Δ ^9^ desaturase C18:0	73.36	73.39	73.33	0.062	0.982	0.728
Elongase	73.75	73.64	73.68	0.045	0.398	0.755
Atherogenic index	0.676	0.644	0.601	0.006	0.077	0.239
Thrombogenic index	0.966	0.972	0.970	0.002	0.416	0.777
Index h:H ^17^	2.247	2.233	2.2440	0.005	0.451	0.399

^1^ Inclusion of BOVAER^®^ (3-nitrooxypropanol, mg/kg DM). ^2^ SEM (standard error of the mean). ^3^ C1 (control vs. BOVAER^®^); C2 (75 vs. 100 mg/kg MS BOVAER^®^). ^4^ Fatty acids from 4 to 14 carbons; ^5^ Fatty acids with more than 16 carbons; ^6^ Saturated fatty acids; ^7^ Unsaturated fatty acids; ^8^ Monounsaturated fatty acids; ^9^ Polyunsaturated fatty acids; ^10^ Odd-chain fatty acids; ^11^ Omega 3 fatty acids; ^12^ Omega 6 fatty acids; ^13^ Total saturated/unsaturated fatty acids ratio; ^14^ Relationship of saturated/unsaturated fatty acids with 18 carbons; ^15^ Relationship of omega 6: omega 3 fatty acids; ^16^ Product/substrate relationship of the enzyme stearoyl-CoA desaturase; ^17^ Relationship of hypocholesterolemic and hypercholesterolemic fatty acids.

## Data Availability

The data presented in this study are available on request from the corresponding author.

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
