# Peer review of "Performance, Meat Quality and Meat Metabolomics Outcomes: Efficacy of 3-Nitrooxypropanol in Feedlot Beef Cattle Diets"

_animals, 2024, doi:10.3390/ani14172576_

Round 1

Reviewer 1 Report

Comments and Suggestions for Authors

Author Response

Comments 1: L122-135, please provide feeding details on feeding amount (% of body weight or kg/head/day).

Response 1: We agree with the correction. We added an explanation about the amount of feed provided: "The animals went through two adaptation periods to adapt to the finishing diet, starting with an intake of 1.5% of the BW. The diets were adjusted daily, based on the reading of trough, allowing the animals a gradual increase in consumption."

We started by providing 1.5% of the animals' body weight, with the aim of adapting the animals to the diet that would be provided, to avoid any disorders or problems related to the diet. The goal was for the animals to consume more than 2% of their body weight with increased consumption, which was achieved throughout the experiment.

Comments 2: L138, please do superscript for m2.

Response 2: We therefore agree to change and place a superscript, modifier from m2 to m2, in L137.

Comments 3: L140, byfeeder or by feeder? Please check.

Response 3: Sorry, we agree with the correction, it was corrected to "By feeders", in 139.

Comments 4: L153, each o animal? What does this mean?

Response 4: Thank you for the consideration! We include this information because the precision platform used measures the weights and gives us information about each animal, individually. Therefore, we were able to obtain a weight gain curve for each animal. Therefore, we were able to monitor the individual performance, by treatment, and overall performance of the animals, throughout the entire experimental period.

To facilitate understanding, I made a change to the sentence: "The slope of these equations could be observed individually per animal, per treatment and obtaining the general average, and accurately represented the total weight gain during the observed period, thus eliminating possible errors attributable to timing or weighing inaccuracies."

Comments 5: L172-173, CH4, H2, CO2→subscript. Please change throughout the manuscript.

Response 5: We agree with the correction, the abbreviations were modified throughout the text.

Comments 6: L196, please check the symbol of degree Celsius throughout the manuscript

Response 6: We agree with the correction, the symbols were changed throughout the text.

Comments 7: L252-261, please details more on what are metabolomic parameters.

Response 7: Thanks for the correction! We added a few lines to explain our objective in performing the metabolomic analysis.

Comments 8: L297-303, please detail more, be sure that other researchers are be able to repeat the procedure with citation.

Response 8: Thank you for your consideration, we agree and improve the writing! We added citations of the procedures that were used to carry out the analyses. The change is found in the subtitle "2.7.3. Metabolite Identification and Quantitation".

Comments 9: L321-332, what techniques did authors use to compare treatment means? Please describe more.

Response 9: We, the authors, apologize and sincerely appreciate the assessment of averages in the statistics section of the manuscript.

Comments 10: Why did not authors analyze orthogonal polynomials for treatment effects? Please explain.

Response 10: The analysis of means was done by orthogonal contrasts, we preferred the contrasts to polynomial regression by adjusting the results curve, considering that only 3 treatments were used; the regression analysis would not have adequate robustness and the results curve would not meet the objectives of the work. In this case, orthogonal contrasts were more indicated, explicitly meeting the objective of the manuscript.

Comments 11: EPM stands for standard error of means is not common, please change to SEM. (Table 02)

Response 11: We agree with the correction, it was changed in all tables.

Comments 12: Please provide p-value for treatment effect. Also, this applies to other tables.

L348: superscript for 3C1.

Response 12: We understand the suggestion of our distinguished colleague, but the C1 contrast performed provides much more robustness and precision to the P value of the treatments. The P value of the treatment only indicates that there are differences but does not indicate where this difference between the treatments would be. In this case, we will not address this suggestion since the analysis we performed is more robust and precise.

Comments 13: Please add interaction study in the section of statistics. (Table 03)

Response 13: We, the authors, thank and apologize for including the statistical model of ingestive behavior (Table 3) in the statistics section.

Comments 14: “Discussion” L501-502, environmental threats related to the use of 3-NOP, what parameters showed a nonenvironmental threats? Please explain.

Response 14: Thanks for the correction! The work was added with the intention of showing that no negative or persistent effects of 3-NOP on soil enzyme activities were observed.

However, we noticed that the quote was loose and contained confusing information in the text. I apologize and noted that it doesn't make much sense for us to use this quote in a work on gas emissions, performance and meat quality, I will remove the quote, if requested, I will post it again.

Reviewer 2 Report

Comments and Suggestions for Authors

The manuscript titled "Optimizing environmental and performance outcomes: Efficacy of 3-Nitrooxypropanol in feedlot beef cattle diets" presents a thorough investigation into the use of 3-nitrooxypropanol (3-NOP) for mitigating methane emissions from beef cattle, while also evaluating its impact on animal performance and meat quality. The study is well-structured and provides comprehensive data, contributing valuable insights to the field of animal science and environmental sustainability.

1. Abstract, please avoid including too many specific statistical details; focus on main findings and significance.

2. The introduction section provides a good background but could benefit from a clearer articulation of the study's objectives and hypotheses. Specifically, the rationale for choosing 75 mg/kg and 100 mg/kg DM doses should be elaborated.

3. Too many studies for 3-NOP, please highlight your study’s novelty.

4. Specify the criteria for animal selection and any exclusion criteria applied. Justify the sample size used in the study and discuss its adequacy for detecting significant differences.

5. L622-L623, please add more discussion. Pyruvate is an intermediate product between the upstream “starch and sucrose metabolism” and the downstream pathway of “glycolysis, suggest to add the reference (doi: 10.1039/d2fo02751h). Also, suggest to add the pathway analysis for the metabolome analysis.

6. Discussion, suggest to compare your findings with more recent literature to strengthen the relevance of your study.

7. Discuss the relevance of the observed changes in meat quality parameters to consumer preferences and marketability.

8. Acknowledge any limitations of the study, such as the short duration or specific conditions under which it was conducted.

Author Response

Comments 1: Abstract, please avoid including too many specific statistical details; focus on main findings and significance.

Response 1: Thanks for the correction! I removed some of the statistical data and left only those of significance (P<.0001), which are the methane emission data.

Comments 2: The introduction section provides a good background but could benefit from a clearer articulation of the study's objectives and hypotheses. Specifically, the rationale for choosing 75 mg/kg and 100 mg/kg DM doses should be elaborated.

Response 2: Thanks for the correction! We added a few lines to explain the use of the two doses, I hope you can understand, if it's not clear I can change the writing again.

Comments 3: Too many studies for 3-NOP, please highlight your study’s novelty.

Response 3: Thank you for the consideration! With the previous consideration (comment 1), I have tried to highlight testing a lower dose of 3-NOP than is generally used. However, the highlight is also the evaluation of meat quality and metabolomics, with the identification and quantification of the metabolites of the meat of these animals.

Not much research was found evaluating the quality of meat from cattle fed with 3-NOP. Likewise, metabolomics of meat from animals that were fed 3-NOP has not been carried out to date.

Comments 4: Specify the criteria for animal selection and any exclusion criteria applied. Justify the sample size used in the study and discuss its adequacy for detecting significant differences.

Response 4: The criteria used to select the sample N were established according to the capacity of the facilities, which, as described in the materials and methods section, can only support 30 animals. For the study in question, since the intake was computed individually at various times of the day (as can be seen in Table 3), ten replicates per treatment provided a precise and robust analysis power for the design in question. The use of individual feeders provided precise and highly reliable analyses for the study in question.

Comments 5: L622-L623, please add more discussion. Pyruvate is an intermediate product between the upstream “starch and sucrose metabolism” and the downstream pathway of “glycolysis”, suggest to add the reference (doi: 10.1039/d2fo02751h). Also, suggest to add the pathway analysis for the metabolome analysis.

Response 5: We, the authors, thank you for including more details about the metabolic pathway of pyruvate in muscle in the discussion of the article.

Comments 6: Discussion, suggest to compare your findings with more recent literature to strengthen the relevance of your study.

Response 6: We appreciate the suggestion of our noble colleague, however, in our view, the articles used are sufficient to support our results.

Comments 7: Discuss the relevance of the observed changes in meat quality parameters to consumer preferences and marketability.

 Response 7: We appreciate the suggestion of our noble colleague, however, it is not the objective of this work to evaluate consumer preferences since we did not carry out a sensory panel. However, we appreciate the suggestion to include this topic in new studies by our groups.

Comments 8: Acknowledge any limitations of the study, such as the short duration or specific conditions under which it was conducted.

Responde 8: Thanks for the comments!

We added the following information "However, more studies will be needed to compare meat quality and metabolomics data, with a larger number of animals, increasing accuracy." at the end of the discussion, recognizing that more experiments evaluating these variables would be necessary, and with a larger experimental number, since our study evaluated only 30 animals, 10 in each treatment.

Reviewer 3 Report

Comments and Suggestions for Authors

This is a good study but the title did not reflect the new findings that is different from previous studies. The meat quality and carcass parameters will help the work to be unique. Please, restructure the title.

"The global meat production has been steadily increasing to meet to meet the growing global demands, with beef emerging prominently" Please, check the statement. 

Table 2 indicated productive performance. What are the productive parameters? 

Comments on the Quality of English Language

Please, check the use of punctation

Some sentences can be restructure to make the message clearer for all readers.

Author Response

-Comments and Suggestions for Authors

Comments 1: This is a good study but the title did not reflect the new findings that is different from previous studies. The meat quality and carcass parameters will help the work to be unique. Please, restructure the title.

Response 1: We appreciate the comment, we have restructured the title.

Comments: "The global meat production has been steadily increasing to meet to meet the growing global demands, with beef emerging prominently" Please, check the statement. 

Response: Thanks for the correction! The information has been corrected.

Comments 2:Table 2 indicated productive performance. What are the productive parameters? 

Response 2: The parameters that indicated productive performance showed a significant effect between animals in the control group and those supplemented with 3-NOP, such as ADG, HCWfinal and ADGcarcass(Average carcass gain), which reflect the animal's gain, and an effect on post-carcass slaughter of these animals. 

-Comments on the Quality of English Language

Comments 3: Please, check the use of punctation. Some sentences can be restructure to make the message clearer for all readers.

Response 3: Thank you for the consideration! We checked the article, changing the use of punctuation in some places to improve understanding for the reader.

Reviewer 4 Report

Comments and Suggestions for Authors

The objective of the current study was to determine the effect of increasing doses of 3 nitrooxypropanol on methane emissions, performance, fatty acid composition, and metabolite concentrations. The impact of 3NOP on fatty acid composition and metabolite concentrations is novel and adds information to the literature about potential impacts and modes of action within animal metabolism.

Specific comments:

Line 66: where cattle “are” subjected

Line 81: a significant “decrease” instead of “effect”

Line 183: whiting? Do you mean “within”?

Line 220-222: more detail needed for EE and ash procedures.

Line 240: need to state which internal standard was used.

Line 309: why is a different comparison made here (control vs 75 and control vs 100) as opposed to the rest of the data (control vs 75 and 75 vs 100). It seems that the comparison should be the same throughout all measures.

Line 331: not typical expression for significance. Suggest P < 0.05.

Table 2: method to determine initial carcass weight and carcass adg are not described in the materials and methods and they need to be included.

Line 357: suggest describing the period effect (current line 366 and line 380 to 381) here and also describing the interaction here (current line 377).

Line 377: BV75ncompared to what? Need to clarify.

Line 380: BV100 and control were not statistically compared. Cannot state that 100 did not differ from con, the analysis was not made. Need to re-word.

Line 447: not worded accurately. 3NOP fed steers did not higher fatty acids, only higher in specific fatty acids. Need to re-word. Suggest: exhibited greater concentrations of C14:1, ….

Line 452-454: What important changes? Need to describe, but Table 7 had no statistical differences.

Line 492: how does it imply a more favorable energetic status? figure 7 needs to be better explained here.

Line 513-514: reduction compared to control? Need to be more specific.

Line 525: remove “higher”

Line 597: lower intramuscular fat is not necessarily beneficial. Consumers prefer the taste of greater intramuscular fat.

Line 604: what figure? figure 6? Need to clarify.

Line 607: “These metabolites” are 3 hydroxybutryate and pyruvate? Need to clarify

Line 618: supplemented with what? 3NOP? Need to clarify.

Line 624-625: suggest: may indicate greater demand for energy to enhance muscle mass and consequently...

Line 626: treated animals from which study? Need to clarify.

Line 628-629: 3NOP supplemented? for which study? need to clarify.

Line 643: clarify that they are 3NOP supplemented animals

Line 652: unclear. An important metabolit “that also differed?” between groups.

Line 656-657. Suggest: which aligns with the greater performance and hot carcass weight in 3NOP supplemented animals in the present study. What performance? Need to be more specific about what “performance” is.

Author Response

Comments 1: Line 66: where cattle “are” subjected

Response 1: Thanks for the correction, corrected to “are”

Comments 2:Line 81: a significant “decrease” instead of “effect”

Response 2 : Thanks for the correction, corrected!

Comments 3:Line 183: whiting? Do you mean “within”?

Response 3: Thanks for the correction, corrected to “within”

Comments 4: Line 220-222: more detail needed for EE and ash procedures.

Response 4: Thanks for the correction, added.

Comments 5: Line 240: need to state which internal standard was used.

Response 5: Thanks for the correction, added.

Comments 6: Line 309: why is a different comparison made here (control vs 75 and control vs 100) as opposed to the rest of the data (control vs 75 and 75 vs 100). It seems that the comparison should be the same throughout all measures.

Response 6 : Thank you for consideration! In metabolomic analyzes we generally perform multivariate analyses, instead of traditional statistics. In this case our objective was not to compare 75 vs. 100 mg/kg of DM, and observe the differences between control vs. 3-NOP, as these are more sensitive analyses. In addition to the analysis of 75 vs. 100 mg/kg DM was not significant for metabolic pathways, so we preferred control vs. 75mg/kg DM and control vs. 100mg/kg of DM, even though it is a little different, the comparison does not escape the objective of the work.

Comments 7: Line 331: not typical expression for significance. Suggest P < 0.05.

Response 7: Thanks for the correction, added.

Comments 8: Table 2: method to determine initial carcass weight and carcass adg are not described in the materials and methods and they need to be included.

Response 8: Thanks for the correction, they were included.

Comments 9: Line 357: suggest describing the period effect (current line 366 and line 380 to 381) here and also describing the interaction here (current line 377).

Response 9: Thanks for the correction, corrected.

Comments 10: Line 377: BV75ncompared to what? Need to clarify.

Response 10: Thanks for the correction, added.

Comments 11: Line 380: BV100 and control were not statistically compared. Cannot state that 100 did not differ from con, the analysis was not made. Need to re-word.

Response 11: Thanks for the correction! I removed the information because even if I reformulated the information it was lost.

Comments 12: Line 447: not worded accurately. 3NOP fed steers did not higher fatty acids, only higher in specific fatty acids. Need to re-word. Suggest: exhibited greater concentrations of C14:1, ….

Response 12: Thanks for the correction! Corrected.

Comments 13: Line 452-454: What important changes? Need to describe, but Table 7 had no statistical differences.

Response 13: Thanks for the correction! We removed it as we stated above which acids had significant effects, and added that table 7 showed no significant effects.

Comments 14: Line 492: how does it imply a more favorable energetic status? figure 7 needs to be better explained here.

Response 14: Thank you for your consideration! In this part of the results, we chose not to explain in detail since we explained in the discussion why this condition is more favorable to animals supplemented with 3-NOP, but we added more explanations in the discussion on this issue.

Comments 15: Line 513-514: reduction compared to control? Need to be more specific.

Response 15: Thanks for the correction! Corrected.

Comments 16: Line 525: remove “higher”

Response 61: Thanks for the correction! Removed.

Comments 17: Line 597: lower intramuscular fat is not necessarily beneficial. Consumers prefer the taste of greater intramuscular fat.

Response 17: Thanks for the correction! I rephrased the quote so it wouldn't be confusing.

Comments 18: Line 604: what figure? figure 6? Need to clarify.

Response 18: Thanks for the correction! Corrected.

Comments 19: Line 607: “These metabolites” are 3 hydroxybutryate and pyruvate? Need to clarify

Response 19: Thanks for the correction! Corrected

Comments 20: Line 618: supplemented with what? 3NOP? Need to clarify.

Response 20: Thanks for the correction! Corrected.

Comments 21: Line 624-625: suggest: may indicate greater demand for energy to enhance muscle mass and consequently...

Response 21: Thanks for the correction! Corrected

Comments 22: Line 626: treated animals from which study? Need to clarify.

Response 22: Thanks for the correction! Corrected

Comments 23: Line 628-629: 3NOP supplemented? for which study? need to clarify.

Response 23: Thanks for the correction! Corrected

Comments 24:Line 643: clarify that they are 3NOP supplemented animals

Response 24: Thanks for the correction! Corrected

Comments 25: Line 652: unclear. An important metabolit “that also differed?” between groups.

Response 25: Thanks for the correction! Corrected

Comments 26: Line 656-657. Suggest: which aligns with the greater performance and hot carcass weight in 3NOP supplemented animals in the present study. What performance? Need to be more specific about what “performance” is.

Response 26: Thanks for the correction! Corrected.
